# OPTIMIZATION DYNAMICS OF EQUIVARIANT AND AUGMENTED NEURAL NETWORKS

## ABSTRACT

We investigate the optimization of multilayer perceptrons on symmetric data. We compare the strategy of constraining the architecture to be equivariant to that of using augmentation. We show that, under natural assumptions on the loss and non-linearities, the sets of equivariant stationary points are identical for the two strategies, and that the set of equivariant layers is invariant under the gradient flow for augmented models. Finally, we show that stationary points may be unstable for augmented training although they are stable for the equivariant models.

## 1  INTRODUCTION

In machine learning, the general goal is to find 'hidden' patterns in data. However, there are sometimes symmetries in the data that are known a priori. Incorporating these manually should, heuristically, reduce the complexity of the learning task. In this paper, we are concerned with training networks, more specifically multilayer perceptrons (MLPs) as defined below, on data exhibiting symmetries that can be formulated as equivariance under a group action. A standard example is the case of translation invariance in image classification.

More specifically, we want to theoretically study the connections between two general approaches to incorporating symmetries in MLPs. The first approach is to construct equivariant models by means of *architectural design*. This framework, known as *Geometric Deep Learning* Bronstein et al. (2017; 2021), exploits the geometric origin of the group $G$ of symmetry transformations by choosing the linear MLP layers and nonlinearities to be equivariant (or invariant) with respect to the action of $G$. In other words, the symmetry transformations commute with each linear (or affine) map in the network, which results in an architecture which manifestly respects the symmetry $G$ of the problem. One prominent example is the spatial weight sharing of convolutional neural networks (CNNs) which are equivariant under translations. Group equivariant convolution networks (GCNNs) Cohen and Welling (2016); Weiler et al. (2018); Kondor and Trivedi (2018) extends this principle to an exact equivariance under more general symmetry groups. The second approach is agnostic to model architecture, and instead attempts to achieve equivariance during training via *data augmentation*, which refers to the process of extending the training to include synthetic samples obtained by subjecting training data to random symmetry transformations.

Both approaches have their benefits and drawbacks. Equivariant models use parameters efficiently through weight sharing along the orbits of the symmetry group, but are difficult to implement and computationally expensive to evaluate in general, since they entail numerical integration over the symmetry group (see, e.g., Kondor and Trivedi (2018)). Data augmentation, on the other hand, is agnostic to the model structure and easy to adapt to different symmetry groups. However, the augmentation strategy is by no means guaranteed to achieve a model which is exactly equivariant: the hope is that model will 'automatically' infer invariances from the data, but there are few theoretical guarantees. Also, augmentation in general entails an inefficient use of parameters and an increase in model complexity and training required.

In this paper, we study equivariant MLPs, which can be characterized by their weights lying in a special subspace $\mathcal{E}$ of the entire parameter space $\mathcal{L}$ (both formally defined below). We use representation theory to compare and analyze the dynamics of gradient flow training of MLPs adhering to either the equivariant or augmented strategy. In particular, we study their stationary points, i.e., the potential limit points of the training, in $\mathcal{E}$. The equivariant models (of course) have optima for their gradient flow there and so are guaranteed to produce a trained model which respects

the symmetry. However, the dynamics under augmented training has previously not been exhaustively explored.

Our main contributions are as follows: First, we provide a technical statement (Lemma 4) about the augmented risk: We show that it can be expressed as the nominal risk averaged over the symmetry group acting on the layers of the MLP. Using this formulation of the augmented risk, we apply standard methods from the analysis of dynamical systems to show:

  (i) The subspace $\mathcal{E}$ is invariant under the gradient flow of the augmented model (Corollary 1). In other words, if the augmented model is equivariantly initialized, it will remain equivariant during training.

 (ii) The set of stationary points in $\mathcal{E}$ for the augmented model is identical to that of the equivariant model (Corollary 2). In other words, compared to the equivariant approach, augmentation introduces no new equivariant stationary points, nor does it exclude existing ones.

(iii) The set of strict local minima in $\mathcal{E}$ for the augmented model is a subset of the corresponding set for the equivariant model. (Proposition 3). In other words, the existence of a stable equivariant minimum is not guaranteed by augmentation.

In addition, we perform experiments on different learning tasks, with different symmetry groups, and discuss the results in the context of our theoretical developments.

## 2    RELATED WORK

The group theory based model for group augmentation we use here is heavily inspired by the framework developed in Chen et al. (2020). Augmentation and manifest invariance/equivariance have been studied from this perspective in a number of papers Lyle et al. (2019; 2020); Mei et al. (2021); Elesedy and Zaidi (2021). More general models for data augmentation have also been considered Dao et al. (2019). Previous work has mostly mostly been concerned with so-called kernel and *feature-averaged* models (see Remark 2 below), and in particular, fully general MLPs as we treat them here have not been considered. The works have furthermore mostly been concerned with proving statistical properties of the models, and not study their dynamic at training. An exception is Lyle et al. (2020), in which it is proven that in linear scenarios, the equivariant models are optimal, but little is known about more involved models.

The dynamics of training *linear equivariant networks* (i.e., MLPs without nonlinearities) has been given some attention in the literature. Linear networks is a simplified, but nonetheless popular theoretical model for analysing neural networks Bah et al. (2022). In (Lawrence et al., 2022), the authors analyse the implicit bias of training a linear neural network with one fully connected layer on top of an equivariant backbone using gradient descent. They also provide some numerical results for non-linear models, but no comparison to data augmentation is made. In Chen and Zhu (2023), completely equivariant linear networks are considered, and an equivalence result between augmentation and restriction is proven for binary classification tasks. However, more realistic MLPs involving non-linearities are not treated at all.

Empirical comparisons of training equivariant and augmented non-equivariant models are common in the literature. Most often, the augmented models are considered as baselines for the evaluation of the equivariant models. More systemic investigations include Gandikota et al. (2021); Müller et al. (2021); Gerken et al. (2022). Compared to previous work, our formulation differs in that the parameter of the augmented and equivariant models are defined on the same vector spaces, which allows us to make a stringent mathematical comparison.

## 3    MATHEMATICAL FRAMEWORK

Let us begin by setting up the framework. We let $X$ and $Y$ be vector spaces and $\mathcal{D}(x, y)$ be a joint distribution on $X \times Y$. We are concerned with training an MLP $\Phi_A : X \to Y$ so that $y \approx \Phi_A(x)$. The MLP has the form

$$x_0 = x, \quad x_{i+1} = \sigma_i(A_i x_i), \quad i \in [L] = \{0, \ldots, L-1\}, \quad \Phi_A(x) = x_L, \tag{1}$$

where $A_i : X_i \to X_{i+1}$ are linear maps (layers) between (hidden) vector spaces $X_i$ with $X = X_0$ and $Y = X_L$, and $\sigma_i : X_{i+1} \to X_{i+1}$ are non-linearities.[1] Note that $A = (A_i)_{i \in [L]}$ parameterizes the network since the non-linearities are assumed to be fixed. We denote the vector space of possible linear layers with $\mathcal{L} = \bigoplus_{i \in [L]} \mathrm{Hom}(X_i, X_{i+1})$, where $\mathrm{Hom}(X_i, X_{i+1})$ is the space of linear maps $X_i \to X_{i+1}$. To train the MLP, we optimize the *nominal risk*

$$R(A) = \mathbb{E}_{\mathcal{D}}(\ell(\Phi_A(x), y)), \tag{2}$$

where $\ell : Y \times Y \to \mathbb{R}$ is a loss function, using gradient descent. In fact, to simplify the analysis, we will mainly study the *gradient flow* of the model, i.e., set the learning rate to be infinitesimal.

### 3.1 REPRESENTATION THEORY AND EQUIVARIANCE

Throughout the paper, we aim to make the MLP *equivariant* towards a group of symmetry transformations of the space $X \times Y$. That is, we consider a group $G$ acting on the vector spaces $X$ and $Y$ through *representations* $\rho_X$ and $\rho_Y$, respectively. A representation $\rho$ of a group $G$ on a vector space $V$ is a map from the group $G$ to the group of invertible linear maps $\mathrm{GL}(V)$ on $V$ that respects the group operation, i.e. $\rho(gh) = \rho(g)\rho(h)$ for all $g, h \in G$. The representation $\rho$ is unitary if $\rho(g)$ is unitary for all $g \in G$. A function $f : X \to Y$ is called equivariant with respect to $G$ if $f \circ \rho_X(g) = \rho_Y(g) \circ f$ for all $g \in G$. We denote the space of equivariant *linear* maps $U \to V$ by $\mathrm{Hom}_G(U, V)$.

Let us recall some important examples that will be used throughout the paper.

**Example 1.** *A simple, but important, representation is the trivial one, $\rho^{\mathrm{triv}}(g) = \mathrm{id}$ for all $g \in G$. If we equip $Y$ with the trivial representation, the equivariant functions $f : X \to Y$ are the invariant ones.*

**Example 2.** $\mathbb{Z}_N^2$ *acts through translations on images $x \in \mathbb{R}^{N,N}$: $(\rho^{\mathrm{tr}}(k, \ell)x)_{i,j} = x_{i-k, j-\ell}$.*

Further examples related to the experiments in subsequent sections are provided in Appendix B.

### 3.2 TRAINING EQUIVARIANT MODELS

In order to obtain a model $\Phi_A$ which respects the symmetries of a group $G$ acting on $X \times Y$, we should incorporate them in our model or training strategy. Note that the distribution $\mathcal{D}(x, y)$ of training data is typically not symmetric in the sense $(x, y) \sim (\rho_X(g)x, \rho_Y(g)y)$. Instead, in the context we consider, symmetries are usually inferred from, e.g., domain knowledge of $X \times Y$. We now formally describe two strategies for training MLPs which respect equivariance under $G$.

**Strategy 1: Manifest equivariance** The first method of enforcing equivariance is to constrain the layers to be manifestly equivariant. That is, we assume that $G$ is acting also on all hidden spaces $X_i$ through representations $\rho_i$, where $\rho_0 = \rho_X$ and $\rho_L = \rho_Y$, and constrain each layer $L_i$ to be equivariant. In other words, we choose the layers $L \in \mathcal{L}$ in the *equivariant subspace*

$$\mathcal{E} = \bigoplus_{i \in [L]} \mathrm{Hom}_G(X_i, X_{i+1}) \tag{3}$$

If we in addition assume that all non-linearities $\sigma_i$ are equivariant, it is straight-forward to show that $\Phi_A$ is *exactly* equivariant under $G$ (see also Lemma 3). We will refer to these models as *equivariant*. The set $\mathcal{E}$ has been extensively studied in the setting of geometric deep learning and explicitly characterized in many important cases Maron et al. (2019a); Cohen et al. (2019); Kondor and Trivedi (2018); Weiler and Cesa (2019); Maron et al. (2019b); Aronsson (2022). In Finzi et al. (2021), a general method for determining $\mathcal{E}$ numerically directly from the $\rho_i$ and the structure of the group $G$ is described.

Defining $\Pi_{\mathcal{E}} : \mathcal{L} \to \mathcal{E}$ as the orthogonal projection onto $\mathcal{E}$, a convenient formulation of the strategy, which is the one we will use, is to optimize the *equivariant risk*

$$R^{\mathrm{eqv}}(A) = R(\Pi_{\mathcal{E}} A). \tag{4}$$

---

[1] Bias terms can also be included via the standard trick to write affine maps as linear – see Appendix F.

**Strategy 2: Data augmentation** The second method we consider is to augment the training data. To this end, we define a new distribution on $X \times Y$ by drawing samples $(x, y)$ from $\mathcal{D}$ and subsequently *augmenting* them by applying the action of a randomly drawn group element $g \in G$ on both data $x$ and label $y$. Training on this augmented distribution can be formulated as optimizing the *augmented risk*

$$R^{\mathrm{aug}}(A) = \int_G \mathbb{E}_{\mathcal{D}}(\ell(\Phi_A(\rho_X(g)x), \rho_Y(g)y)) \, \mathrm{d}\mu(g) \tag{5}$$

Here, $\mu$ is the (normalised) *Haar* measure on the group Krantz and Parks (2008), which is defined through its invariance with respect to the action of $G$ on itself; if $h$ is distributed according to $\mu$ then so is $gh$ for all $g \in G$. This property of the Haar measure will be crucial in our analysis. Choosing another measure would cause the augmentation to be biased towards certain group elements, and is not considered here. Note that if the data $\mathcal{D}$ already is symmetric in the sense that $(x, y) \sim (\rho_X(g)x, \rho_Y(g)y)$, the augmentation acts trivially.

**Remark 1.** *(5) is a simplification – in practice, the actual function that is optimized is an empirical approximation of $R^{\mathrm{aug}}$ formed by sampling of the group $G$. Our results are hence about an 'infinite-augmentation limit' that still should have high relevance at least in the 'high-augmentation region' due to the law of large numbers. To properly analyse this transition carefully is important, but beyond the scope of this work.*

In our analysis, we want to compare the two strategies when training the same model. We will make three global assumptions.

**Assumption 1.** *The group $G$ is acting on all hidden spaces $X_i$ through unitary representations $\rho_i$.*

**Assumption 2.** *The non-linearities $\sigma_i : X_{i+1} \to X_{i+1}$ are equivariant.*

**Assumption 3.** *The loss $\ell$ is invariant, i.e. $\ell(\rho_Y(g)y, \rho_Y(g)y') = \ell(y, y')$, $y, y' \in Y$, $g \in G$.*

Let us briefly comment on these assumptions. The first assumption is needed for the restriction strategy to be well defined. The technical part – the unitarity – is not a true restriction: As long as all $X_i$ are finite-dimensional and $G$ is compact, we can redefine the inner products on $X_i$ to ensure that all $\rho_i$ become unitary. The second assumption is required for the equivariant strategy to be sound – if the $\sigma_i$ are not equivariant, they will explicitly break equivariance of $\Phi_A$ even if $A \in \mathcal{E}$. The third assumption guarantees that the loss-landscape is 'unbiased' towards group transformations, which is certainly required to train any model respecting the symmetry group.

We also note that all assumptions are in many settings quite weak – we already commented on Assumption 1. As for Assumption 2, note that e.g. any non-linearity acting pixel-wise on an image will be equivariant to both translations and rotations by multiples of $\pi/2$. In the same way, any loss comparing images pixel by pixel will be, so that Assumption 3 is satisfied. Furthermore, if we are trying to learn an invariant function the final representation $\rho_Y$ is trivial and Assumption 3 is trivially satisfied.

**Remark 2.** *Before proceeding, let us mention a different strategy to build equivariant models:* feature averaging *Lyle et al. (2019). This strategy refers to altering the model by averaging it over the group:*

$$\Phi_A^{\mathrm{FA}}(x) := \int_G \rho_Y(g)^{-1} \Phi_A(\rho_X(g)x) \, \mathrm{d}\mu(g). \tag{6}$$

*In words, the value of the feature averaged network at a datapoint $x$ is obtained by calculating the outputs of the original model $\Phi_A$ on transformed versions of $x$, and averaging the re-adjusted outputs over $G$. Note that the modification of an MLP here does not rely on explicitly controlling the weights. It is not hard to prove (see e.g. (Lyle et al., 2020, Prop. 2)) that under the invariance assumption on $\ell$,*

$$R^{\mathrm{aug}}(A) = \mathbb{E}(\ell(\Phi_A^{\mathrm{FA}}(x), y)). \tag{7}$$

### 3.3 LIFTED REPRESENTATIONS AND THEIR PROPERTIES

We can lift the representations $\rho_i$ to representations $\overline{\rho}_i$ of $G$ on $\mathrm{Hom}(X_i, X_{i+1})$ through

$$\overline{\rho}_i(g)A_i = \rho_{i+1}(g)A_i\rho_i(g)^{-1}, \tag{8}$$

and from that derive a representation $\overline{\rho}$ on $\mathcal{L}$ according to $(\overline{\rho}(g)A)_i = \overline{\rho}_i(g)A_i$. Since the $\rho_i$ are unitary, with respect to the appropriate canonical inner products, so are $\overline{\rho}_i$ and $\overline{\rho}$.

Before proceeding we establish some simple, but crucial facts, concerning the lifted representation $\overline{\rho}$ and the way it appears in the general framework. We will need the following two well-known lemmas. Proofs are presented in Appendix A.

**Lemma 1.** $A \in \mathcal{E}$ if and only if $\overline{\rho}(g)A = A$ for all $g \in G$.

**Lemma 2.** For any $A \in \mathcal{L}$ the orthogonal projection $\Pi_{\mathcal{E}}$ is given by

$$\Pi_{\mathcal{E}} A = \int_G \overline{\rho}(g) A \, \mathrm{d}\mu(g). \tag{9}$$

We now prove a relation between transforming the input and transforming the layers of an MLP.

**Lemma 3.** Under Assumption 2, for any $A \in \mathcal{L}$ and $g \in G$ we have

$$\Phi_A(\rho_X(g)x) = \rho_Y(g)\Phi_{\overline{\rho}(g)^{-1}A}(x). \tag{10}$$

In particular, $\Phi_A$ is equivariant for every $A \in \mathcal{E}$.

*Proof.* The in particular part follows from $\overline{\rho}(g)^{-1}A = A$ for $A \in \mathcal{E}$. To prove the main statement, we use the notation (1): $x_i$ denotes the outputs of each layer of $\Phi_A$ when it acts on the input $x \in X$. Also, for $g \in G$, let $x_i^g$ denote the outputs of each layer of the network $\Phi_{\overline{\rho}(g)^{-1}A}$ when acting on the input $\rho_X(g)^{-1}x$. If we show that $\rho_i(g)x_i^g = x_i$ for $i = [L+1]$, the claim follows. We do so via induction. The case $i = 0$ is clear: $\rho_X(g)x_0^g = \rho_X(g)\rho_X(g)^{-1}x = x = x_0$. As for the induction step, we have

$$\rho_{i+1}(g)x_{i+1}^g = \rho_{i+1}(g)\sigma_i(\overline{\rho}_i(g)^{-1}A_i x_i^g) = \rho_{i+1}(g)\sigma_i(\rho_{i+1}(g)^{-1}A_i \rho_i(g)x_i^g) \tag{11}$$
$$= \sigma_i(\rho_{i+1}(g)\rho_{i+1}(g)^{-1}A_i \rho_i(g)x_i^g) = \sigma_i(A_i \rho_i(g)x_i^g) = \sigma_i(Ax_i) = x_{i+1},$$

where in the second step, we have used the definition of $\overline{\rho}_i$, in the third, Assumption (2), and the fifth step follows from the induction assumption. $\square$

The above formula has an immediate consequence for the augmented loss.

**Lemma 4.** Under Assumptions 2 and 3, the augmented risk can be expressed as

$$R^{\mathrm{aug}}(A) = \int_G R(\overline{\rho}(g)A) \, \mathrm{d}\mu(g). \tag{12}$$

*Proof.* From Lemma 3 and Assumption (3), it follows that for any $g \in G$ we have $\ell(\Phi_{\overline{\rho}(g)^{-1}A}(x), y) = \ell(\rho_Y(g)^{-1}\Phi_A(\rho_X(g)x), y) = \ell(\Phi_A(\rho_X(g)x), \rho_Y(g)y)$. Taking the expectation with respect to the distribution $\mathcal{D}$, and then integrating over $g \in G$ yields

$$R^{\mathrm{aug}}(A) = \int_G R(\overline{\rho}(g)^{-1}A) \, \mathrm{d}\mu(g) = \int_G R(\overline{\rho}(g^{-1})A) \, \mathrm{d}\mu(g). \tag{13}$$

Using the fact that the Haar measure is invariant under inversion proves the statement. $\square$

We note the likeness of (12) to (7): In both equations, we are averaging risks of transformed models over the group. However, in (7), we average over transformations of the *input data*, whereas in (12), we average over transformations of the *layers*. The latter fact is crucial – it will allow us analyse the dynamics of gradient flow.

Before moving on, let us introduce one more notion. When considering the dynamics of training we will encounter elements of the tensor product $\mathcal{L} \otimes \mathcal{L}$, which also carries a representation $\overline{\rho}^{\otimes 2}$ of $G$ lifted by $\overline{\rho}$ according to

$$\overline{\rho}^{\otimes 2}(g)(A \otimes B) = (\overline{\rho}(g)A) \otimes (\overline{\rho}(g)B). \tag{14}$$

We refer to the vector space of elements invariant under this action as $\mathcal{E}^{\otimes 2}$, and the orthogonal projection onto it as $\Pi_{\mathcal{E}^{\otimes 2}}$. As for $\mathcal{L}$, the induced representation on $\mathcal{L} \otimes \mathcal{L}$ can be used to express the orthogonal projection.

**Lemma 5.** For any $A, B \in \mathcal{L}$ the orthogonal projection $\Pi_{\mathcal{E}^{\otimes 2}}$ is given by

$$\Pi_{\mathcal{E}^{\otimes 2}}(A \otimes B) = \int_G \overline{\rho}^{\otimes 2}(g)(A \otimes B) \, \mathrm{d}\mu(g). \tag{15}$$

The space $\mathcal{E}^{\otimes 2}$ consists bilinear forms on $\mathcal{L}$. Importantly, if we represent them as matrices with respect to an ONB which can be subdivided into an ONB of $\mathcal{E}$ and one of $\mathcal{E}^\perp$, it will be block diagonal. This can be formulated as follows:

**Lemma 6.** *For any $M \in \mathcal{L} \otimes \mathcal{L}$, $A \in \mathcal{E}$ and $B \in \mathcal{E}^\perp$ we have*

$$(i) \ (\Pi_{\mathcal{E}^{\otimes 2}} M)\,[A, A] = M[A, A]\,, \tag{16}$$

$$(ii) \ (\Pi_{\mathcal{E}^{\otimes 2}} M)\,[A, B] = 0\,. \tag{17}$$

## 4 DYNAMICS OF THE GRADIENT FLOW

We have now established the framework of optimization for symmetric models that we need to investigate the gradient flow for the equivariant and augmented models. In particular, we want to compare the gradient flow dynamics of the two models as it pertains to the equivariance with respect to the symmetry group $G$ during training. To this end, we consider the gradient flows of the nominal, equivariant and augmented risks

$$\dot{A} = -\nabla R(A)\,, \quad \dot{A} = -\nabla R^{\mathrm{eqv}}(A)\,, \quad \dot{A} = -\nabla R^{\mathrm{aug}}(A)\,, \quad A \in \mathcal{L}\,. \tag{18}$$

### 4.1 EQUIVARIANT STATIONARY POINTS

We first establish the relation between the gradients of the equivariant and augmented models for an initial condition $A \in \mathcal{E}$.

**Proposition 1.** *For $A \in \mathcal{E}$ we have $\nabla R^{\mathrm{aug}}(A) = \Pi_{\mathcal{E}} \nabla R(A) = \nabla R^{\mathrm{eqv}}(A)$.*

*Proof.* Taking the derivative of (12) the chain rule yields

$$\langle \nabla R^{\mathrm{aug}}(A), B \rangle = \int_G \langle \overline{\rho}(g)^{-1} \nabla R(\overline{\rho}(g)A), B \rangle \, \mathrm{d}\mu(g) = \int_G \langle \nabla R(\overline{\rho}(g)A), \overline{\rho}(g)B \rangle \, \mathrm{d}\mu(g)\,, \tag{19}$$

where $B \in \mathcal{L}$ is arbitrary and we have used the unitarity of $\overline{\rho}$. Using that $\overline{\rho}(g)A = A$ for every $A \in \mathcal{E}$ (Lemma 2), we see that the last integral equals

$$\int_G \langle \nabla R(A), \overline{\rho}(g)B \rangle \, \mathrm{d}\mu(g) = \langle \nabla R(A), \Pi_{\mathcal{E}} B \rangle = \langle \Pi_{\mathcal{E}} \nabla R(A), B \rangle\,, \tag{20}$$

where we have used orthogonality of $\Pi_{\mathcal{E}}$ in the final step, which establishes the first equality of the proposition.

The second equality follows immediately from the chain rule applied to (4)

$$\nabla R^{\mathrm{eqv}}(A) = \Pi_{\mathcal{E}} \nabla R(\Pi_{\mathcal{E}} A) = \Pi_{\mathcal{E}} \nabla R(A)\,, \tag{21}$$

where we have used that $\Pi_{\mathcal{E}}$ is self-adjoint and $\Pi_{\mathcal{E}} A = A$ for every $A \in \mathcal{E}$ in the last step. $\square$

A direct consequence of Proposition 1 is that for any $A \in \mathcal{E}$ we have $\nabla R^{\mathrm{aug}}(A) \in \mathcal{E}$ for the gradient, which establishes the following important result.

**Corollary 1.** *The equivariant subspace $\mathcal{E}$ is invariant under the gradient flow of $R^{\mathrm{aug}}$.*

A further immediate consequence of Proposition 1 is the fact that if the initialization of the networks is equivariant, the gradient flow dynamics of the augmented and equivariant models will be identical. In particular, we have the following result.

**Corollary 2.** *$A^* \in \mathcal{E}$ is a stationary point of the gradient flow of $R^{\mathrm{aug}}$ if and only if it is a stationary point of the gradient flow of $R^{\mathrm{eqv}}$.*

### 4.2 STABILITY OF THE EQUIVARIANT STATIONARY POINTS

We now consider the stability of equivariant stationary points, and more generally of the equivariant subspace $\mathcal{E}$, under the augmented gradient flow of $R^{\mathrm{aug}}$. Of course, $\mathcal{E}$ is manifestly stable under the equivariant gradient flow of $R^{\mathrm{eqv}}$. To make statements about the stability we establish the connection between the Hessians of $R$, $R^{\mathrm{eqv}}$ and $R^{\mathrm{aug}}$, which can be considered as bilinear forms on $\mathcal{L}$, i.e. as elements of the tensor product space $\mathcal{L} \otimes \mathcal{L}$.

**Proposition 2.** *For $A \in \mathcal{E}$ we have $\nabla^2 R^{\mathrm{aug}}(A) = \Pi_{\mathcal{E} \otimes 2} \nabla^2 R(A)$ and $\nabla^2 R^{\mathrm{eqv}}(A) = \Pi_{\mathcal{E}}^{\otimes 2} \nabla^2 R(A)$.*

*Proof.* Taking the second derivative of (12) yields

$$\nabla^2 R^{\mathrm{aug}}(A)[B,C] = \int_G \nabla^2 R(\overline{\rho}(g)A)[\overline{\rho}(g)B, \overline{\rho}(g)C]\,\mathrm{d}\mu(g) = \int_G \overline{\rho}^{\otimes 2}(g)\nabla^2 R(A)[B,C]\,\mathrm{d}\mu(g)\,, \tag{22}$$

where $B, C \in \mathcal{L}$ are arbitrary and we have used $\overline{\rho}(g)A = A$ for $g \in G$ and $A \in \mathcal{E}$ and the definition of $\overline{\rho}^{\otimes 2}$. Lemma 5 then yields the first equality.

The second statement again follows directly from the chain rule twice applied to (4)

$$\nabla^2 R^{\mathrm{eqv}}(A) = \Pi_{\mathcal{E}}^{\otimes 2} \nabla^2 R(\Pi_{\mathcal{E}} A) = \Pi_{\mathcal{E}}^{\otimes 2} \nabla^2 R(A)\,, \tag{23}$$

where we have used $\Pi_{\mathcal{E}} A = A$ for $A \in \mathcal{E}$ and the fact that $\Pi_{\mathcal{E}}$ is self-adjoint. $\square$

**Proposition 3.** *For $A^* \in \mathcal{E}$ the following implications hold:*

*i) If $A^*$ is a strict local minimum of $R$, it is a strict local minimum of $R^{\mathrm{aug}}$.*

*ii) If $A^*$ is a strict local minimum of $R^{\mathrm{aug}}$, it is a strict local minimum of $R^{\mathrm{eqv}}$.*

*Proof. i)* Assume $\nabla R(A^*) = 0$ and $\nabla^2 R(A^*)$ positive definite. From Proposition 1 we then have $\nabla R^{\mathrm{aug}}(A^*) = \Pi_{\mathcal{E}} \nabla R(A^*) = 0$. Furthermore, for $B \neq 0$ Proposition 2 implies

$$\nabla^2 R^{\mathrm{aug}}(A^*)[B,B] = \Pi_{\mathcal{E} \otimes 2} \nabla^2 R(A^*)[B,B] = \int_G \nabla^2 R(A^*)[\overline{\rho}(g)B, \overline{\rho}(g)B]\,\mathrm{d}\mu(g) > 0\,, \tag{24}$$

where in the last step we have used the fact that the integrand is positive since $\overline{\rho}(g)B \neq 0$ for any $g \in G$ and $B \neq 0$, and $\nabla^2 R(A^*)[B,B] > 0$ for $B \neq 0$.

*ii)* Assume $\nabla R^{\mathrm{aug}}(A^*) = 0$ and $\nabla^2 R^{\mathrm{aug}}(A^*)$ positive definite. From Proposition 1 we have $\nabla R^{\mathrm{eqv}}(A^*) = \nabla R^{\mathrm{aug}}(A^*) = 0$. Proposition 2 implies that $\nabla^2 R^{\mathrm{eqv}}(A^*)[B,B]$ equals

$$\nabla^2 R(A^*)[\Pi_{\mathcal{E}} B, \Pi_{\mathcal{E}} B] = \Pi_{\mathcal{E} \otimes 2} \nabla^2 R(A^*)[\Pi_{\mathcal{E}} B, \Pi_{\mathcal{E}} B] = \nabla^2 R^{\mathrm{aug}}(A^*)[\Pi_{\mathcal{E}} B, \Pi_{\mathcal{E}} B]\,, \tag{25}$$

where we used that $\Pi_{\mathcal{E}} B \in \mathcal{E}$, together with the first part of Lemma 6. Consequently, $\nabla^2 R^{\mathrm{eqv}}(A^*)[B,B] > 0$ for $\Pi_{\mathcal{E}} B \neq 0$ which completes the proof. $\square$

The converse of Proposition 3 is not true. A a concrete counterexample is provided in Appendix C.

Finally, let us remark an interesting property of the dynamics of the augmented gradient flow of $R^{\mathrm{aug}}$ near $\mathcal{E}$. Decomposing $A$ near $\mathcal{E}$ as $A = x + y$, with $x \in \mathcal{E}$ and $y \in \mathcal{E}^{\perp}$, with $y \approx 0$, and linearising in the deviation $y$ from the equivariant subspace $\mathcal{E}$ yields

$$\dot{x} + \dot{y} = -\nabla R^{\mathrm{aug}}(x) - y^* \nabla^2 R^{\mathrm{aug}}(x) + \mathcal{O}(\|y\|^2). \tag{26}$$

From Proposition 1 we have $\nabla R^{\mathrm{aug}}(x) \in \mathcal{E}$, and Proposition 2 (ii) together with Lemma 6 implies that $y^* \nabla^2 R^{\mathrm{aug}}(x) \in \mathcal{E}^{\perp}$. Consequently, the dynamics approximately decouple:

$$\begin{cases} \dot{x} &= -\nabla R^{\mathrm{aug}}(x) &+ \mathcal{O}(\|y\|^2) \\ \dot{y} &= -y^* \nabla^2 R^{\mathrm{aug}}(x) &+ \mathcal{O}(\|y\|^2) \end{cases}. \tag{27}$$

We observe that Proposition 1 now implies that the dynamics restricted to $\mathcal{E}$ is identical to that of the equivariant gradient flow, and that the stability of $\mathcal{E}$ is completely determined by the spectrum of $\nabla^2 R^{\mathrm{aug}}$ restricted to $\mathcal{E}^{\perp}$. Furthermore, as long the parameters are close to $\mathcal{E}$, the dynamics of the part of $A$ in $\mathcal{E}$ for the two models are almost equal.

In terms of training augmented models with equivariant initialization, these observations imply that while the augmented gradient flow restricted to $\mathcal{E}$ will converge to the local minima of the corresponding equivariant model, it may diverge from the equivariant subspace $\mathcal{E}$ due to noise and numerical errors as soon as $\nabla^2 R^{\mathrm{aug}}(x)$ restricted to $\mathcal{E}^{\perp}$ has negative eigenvalues. This could potentially be mitigated by introducing a penalty proportional to $\|A_{\mathcal{E}^{\perp}}\|^2$ in the augmented risk. We leave it to future work to analyse this further.

## 5 EXPERIMENTS

We perform some simple experiments to study the dynamics of the gradient flow *near the equivariant subspace* $\mathcal{E}$. From our theoretical results, we expect the following.

(i) The set $\mathcal{E}$ is invariant, but not necessarily stable, under the gradient flow of $R^{\mathrm{aug}}$.

(ii) The dynamics in $\mathcal{E}^\perp$ for $R^{\mathrm{aug}}$ should (initially) be much 'slower' than in $\mathcal{E}$, in particular compared to the nominal gradient flow of $R$.

We consider three different learning tasks, with different symmetry groups and data sets:

PERM  Permutation invariant graph classification (using small synthetically generated graphs)

TRANS  Translation invariant image classification (on a subsampled version of MNIST Lecun et al. (1998))

ROT  Rotation equivariant image segmentation (on synthetic images of simple shapes).

The general setup is as follows: We consider a group $G$ acting on vector spaces $(X_i)_{i=0}^L$. We construct a multilayered perceptron $\Phi_A : X_0 \to X_L$ as above. The non-linearities are always chosen as non-linear functions $\mathbb{R} \to \mathbb{R}$ applied elementwise, and are therefore equivariant under the actions we consider. To mitigate the vanishing gradient problem, we use layer normalization Ba et al. (2016) which can be accommodated as an equivariant nonlinearity $\sigma_i$. We build our models with PyTorch Paszke et al. (2019). Detailed descriptions of e.g. the choice of hidden spaces, non-linearities and data, are provided in Appendix D. In the interest of reproducibility, we also provide the code in the supplementary material.

We initialize $\Phi_A$ with equivariant layers $A^0 \in \mathcal{E}$ by drawing matrices randomly from a standard Gaussian distribution, and then projecting them orthogonally onto $\mathcal{E}$. We train the network on (finite) datasets $\mathcal{D}$ using gradient descent in three different ways.

NOM  A gradient descent, with gradient accumulated over the entire dataset (to emulate the 'non-empirical' risk $R$ as we have defined it here): Data is fed forward through the MLP in mini-batches as usual, but gradients are calculated by taking the averages over mini-batches.

AUG  As NOM , but with $N^{\mathrm{aug}}$ passes over data where each mini-batch is augmented using a randomly sampled group element $g \sim \mu$. The gradient is again averaged over all passes, to model the augmented risk $R^{\mathrm{aug}}$ as closely as possible.

EQUI  As NOM , but the gradient is projected onto $\mathcal{E}$ before the gradient step is taken. This corresponds to the equivariant risk $R^{\mathrm{eqv}}$ and produces networks which are manifestly equivariant.

The learning rate $\tau$ is equal to $5 \cdot 10^{-5}$ in all three experiments. In the limit $\tau \to 0$ and $N^{\mathrm{aug}} \to \infty$, this exactly corresponds to letting the layers evolve according to gradient flow with respect to $R$, $R^{\mathrm{aug}}$ and $R^{\mathrm{eqv}}$, respectively. For each task, we train the networks for 50 epochs. After each epoch we record $\|A - A^0\|$, i.e. the distance from the starting position $A^0$, and $\|A_{\mathcal{E}^\perp}\|$, i.e. the distance from $\mathcal{E}$ or equivalently the 'non-equivariance'. Each experiment is repeated 30 times, with random initialisations.

### 5.1 RESULTS

In Figure 1, we plot the evolution of the values $\|A_{\mathcal{E}^\perp}\|$ against the evolution of $\|A - A^0\|$. The opaque line in each plot is formed by the average values for all thirty runs, whereas the fainter lines are the 30 individual runs.

In short, the experiments are consistent with our theoretical results. In particular, we observe that the equivariant submanifold is consistently unstable (i) in our repeated augmented experiments. In PERM and TRANS, we also observe the hypothesized 'stabilising effect' (ii) on the equivariant subspace: the AUG model stays much closer to $\mathcal{E}$ than the NOM model – the shift orthogonal to $\mathcal{E}$ is smaller by several orders of magnitude. For ROT, the AUG and NOM models are much closer to each other, but note that also here, the actual shifts orthogonal to $\mathcal{E}$ are very small compared to the total shifts $\|A - A^0\|$ – on the order $10^{-7}$ compared to $10^{-3}$.

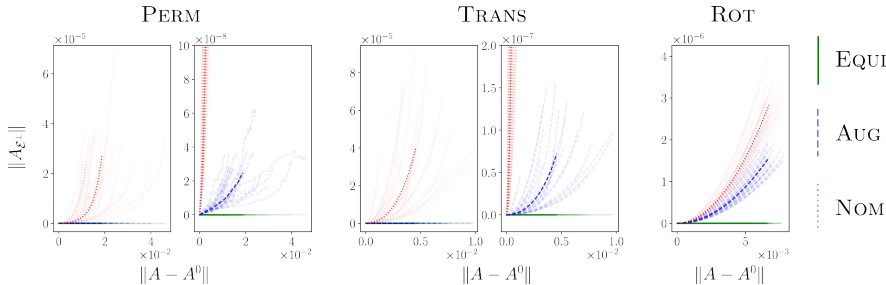

Figure 1: Results of the experiments: $\|A_{\mathcal{E}\perp}\|$ is plotted versus $\|A - A^0\|$. Opaque lines are to mean values, transparent lines are individual experiments. For PERM and TRANS , the two plots depict the same data with different scales on the $\|A_{\mathcal{E}\perp}\|$-axis.

The reason for the different behaviour in the rotation experiment cannot be deduced from our theory. Our hypothesis is that it is the different symmetry groups, or rather different spaces $\mathcal{E}$, plays a role here. Proposition 1 tells us that the difference between gradients of the NOM and AUG models are given by the component orthogonal to $\mathcal{E}$ of $\nabla R$. Hence, if $\mathcal{E}$ is of low dimension, it can be assumed that a lot of the gradient energy disappears. A good proxy for the size is the relation between $\dim \mathcal{E}$ and $\dim \mathcal{L}$. In Appendix E.1, we calculate these fractions and find it to be much larger for ROT than in the other experiments. This is in agreement with the augmented ROT model staying closer to its NOM counterpart than the other cases. In Appendix E.2, we investigate this further (empirically) by repeating the TRANS experiment for other groups. The trend continues: The lower $\dim \mathcal{E} / \dim \mathcal{L}$, the closer the augmented model stays to $\mathcal{E}$. Needless to say, this argument is purely heuristic. Finding a proper theoretical explaination for this is however beyond the scope of this paper.

## 6 CONCLUSION

In this paper we investigated the dynamics of gradient descent for augmented and equivariant models, and how they are related. In particular, we showed that the models have the same set of equivariant stationary points, but that the stability of these points may differ. Furthermore, when initialized to the equivariant subspace $\mathcal{E}$, the dynamics of the augmented model is identical to that of the equivariant one. In a first order approximation, dynamics on $\mathcal{E}$ and $\mathcal{E}^\perp$ even decouple for the augmented models.

These findings have important practical implications for the two strategies for incorporating symmetries in the learning problem. The fact that their equivariant stationary points agree implies that there are no equivariant configurations that cannot be found using manifest equivariance. Hence, the more efficient parametrisation of the equivariant models neither introduces nor excludes equivariant stationary points compared to the less restrictive augmented approach. Conversely, if we can control the potential instability of the non-equivariant subspace $\mathcal{E}^\perp$ in the augmented gradient flow, it will find the same equivariant minima as its equivariant counterpart. One way to accomplish the latter would be to introduce a penalty proportional to $\|A_{\mathcal{E}\perp}\|^2$ in the augmented risk.

This work is a first step towards properly understanding the effects of augmentation on the dynamics of gradient flows, but more research is needed. For instance, although we showed that the dynamics *on* $\mathcal{E}$ is identical for the augmented and equivariant models, and understand their behaviour *near* $\mathcal{E}$, our results say nothing about the dynamics *away from* $\mathcal{E}$ for the augmented model. Indeed, there is nothing stopping the augmented gradient flow from leaving $\mathcal{E}$ – although initialized very near it – or from coming back again. To analyse the global properties of the augmented gradient flow, in particular to calculate the spectra of $\nabla^2 R^{\mathrm{aug}}$ in concrete cases of interest, is an important direction for future research.

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
