# OpenReview forum: "Optimization Dynamics of Equivariant and Augmented Neural Networks"
_ICLR.cc/2024/Conference — ICLR 2024 Conference Withdrawn Submission_

### Official Review · Reviewer_t5ps · 2023-10-26

**Soundness:** 3 good
**Presentation:** 4 excellent
**Contribution:** 3 good
**Rating:** 5
**Confidence:** 4

**Summary:**

This paper is well-written and accessible but the result seems small — I cannot really assess how significant the result is.

The paper considers the situation when the problem at hand has a symmetry which we would like to introduce into our learned model. It compares two conventional ways of imposing symmetries. The first is manifesting equivariance, i.e. projecting linear layers into the linear space of equivariant maps. The second is data augmentation. The two ways leads to different loss functions; how the optimization dynamics is different for them?

With some simple algebraic manipulations, the paper demonstrates that (1) the equivariant subspace of network's linear maps is invariant under gradient flow of augmented loss, and (2) the augmented and equivariant losses have the same stationary points on the equivariant subspace.

However, as the paper demonstrates theoretically and empirically, these stationary points might become unstable for the augmented loss, meaning that while seemingly similar, the two ways of imposing symmetries might lead to different training outcomes.

The paper is well-written and well-structured. The literature review seems decent. The experiments are illustrative, though simple.

As a theoretical weakness (well-mentioned by the authors), the paper tells nothing about stationary points out of the equivariant subspace. In usual practice, linear layers are initialized in a standard way, not at the equivariant space, even when the dataset is augmented. It is an interesting question, whether local minima out of the equivariant space exist.

**Strengths:**

The paper is clearly written, the problem is quite interesting, the theoretical contributions are certainly correct, there are some interesting numerical observations, this may ultimately lead to exciting results.

**Weaknesses:**

The results are not striking, though not entirely trivial. Obviously, one would like to see more exciting examples to justify the whole formalism (though it's obviously hard to produce new equivariant architectures).

**Questions:**

Can we do a progressive hybrid between data augmentation and symmetries of the network (e.g. progressively remove symmetries and augment the data)?

---

### Official Review · Reviewer_fUBC · 2023-11-01

**Soundness:** 3 good
**Presentation:** 3 good
**Contribution:** 2 fair
**Rating:** 5
**Confidence:** 4

**Summary:**

The paper analyzes two strategies of training equivariant neural networks: direct symmetrization of the model in each layer and symmetrizing ("augmenting") the risk. The main theoretical results characterize and compare stationary points and local minima of the resulting loss surfaces when restricted to the subspace $\mathcal E$ of symmetrized models. In particular, Proposition 1 shows that the gradients of both risks on $\mathcal E$ are given by projecting the gradient of the non-symmetric risk to $\mathcal E$; Proposition 2 describes the Hessians of the two risks; and Proposition 3 states an inclusion hierarchy for strict local minima of different risks. Also, the paper illustrates the obtained theoretical results with several gradient descent experiments; in particular, shows that the symmetric subspace $\mathcal E$ can be unstable under the GD with symmetrized risk, but not as much as with unsymmetrized risk.

**Strengths:**

The paper is well-written. I could follow it without difficulty and I think that it is accessible to a wide machine learning audience with little or no background in representation theory. I have not spotted any mistakes except for the one in Remark 2 (see Weaknesses below).

The paper contributes some new simple, but general and rigorous results that shed light on different strategies of constructing equivariant models, and on relations between these strategies. I have not seen these results before.

Though the paper is mostly theoretical, it includes some experiments illustrating the theoretical results.

**Weaknesses:**

**Significance.** I find the results interesting, but 'm not convinced that they are significant enough for an ICLR paper.
1. The main text contains a large number of fairly elementary lemmas followed by Propositions 1-3. The latter are presumably the main results of the paper, and are also relatively elementary. The most important conclusion seems to be that the subspace $\mathcal E$ of models symmetrized in each layer is invariant under the GD with the augmented risk (moreover, this dynamics coincides with the dynamics of the directly symmetrized models). It is claimed that this observation is important, but it is only discussed very briefly why. The provided experiments do not show that it is useful for any practical purpose.
2. Another observation of the paper is that the invariant subspace $\mathcal E$ can be unstable and that the instability can be studied by decoupling and linearization. However, this observation as such is not very informative, and the paper does not prove any specific results here.
3. In Remark 2 the paper mentions "feature averaging", which is the symmetrization of the overall model rather than its individual layers. This symmetrization strategy is more widely applicable in practice thanks to its generality, ease of use and because it reliably improves model performance. The respective equivariant models form a larger subspace than the subspace $\mathcal E$ of the layerwise equivariant models considered in the paper. The problem with the layerwise equivariant models $\mathcal E$ is that they may be not expressive enough to fit the data well. Remark 2 attempts to put the "feature averaged" models into the context of the present work, but it appears to me that the statement made there is not correct (see below), so the connection between the "feature averaged" models and the results of the paper remains unclear to me.

**Wrong or unclear statements.** I don't see how Eq. (7) in Remark 2 can be true. Feature averaging as defined in (6) averages the models, while the augmented risk averages the risk. Specifically, suppose, for example, that $X=Y=\mathbb R$, the data set $\mathcal D$ consists of a single point $(1,y_*)$, the group $G=\mathbb Z_2=\\{e, g\\}$ acts nontrivially in $X$ (by $\rho_X(g)(x)=-x$) and trivially in $Y$ (by $\rho_Y(g)(y)=y$). Suppose the loss is standard quadratic. Then the l.h.s. of (7) is $\tfrac{1}{2}((\Phi_A(1)-y_*)^2+(\Phi_A(-1)-y_*)^2),$ while the r.h.s. is $(\tfrac{1}{2}(\Phi_A(1)+\Phi_A(-1))-y_*)^2,$ and is generally different.

**Minor issues.**
1. It seems that in Proposition 3, statement ii), the paper implicitly assumes that the equivariant risk $R^{\mathrm{eqv}}$ is defined not on the whole space $\mathcal L$ of all models, but rather on its symmetrized subspace $\mathcal E$ (this is used in the end of the proof, and the statement seems to be wrong otherwise). It should be described more carefully on which space each of the risks is considered.
2. The paper mentions that the representations are assumed to be *unitary* - does that mean *orthogonal* in the present context? The paper does not specify the underlying field, but it seems to be implicitly assumed everywhere that it is $\mathbb R$.

**Questions:**

At present, the bulk of the paper (pages 1-5) is filled with various preliminaries. I think that the paper would look much stronger if it included more discussion and/or a deeper analysis of its main results.

---

### Official Review · Reviewer_kDvS · 2023-11-01

**Soundness:** 3 good
**Presentation:** 3 good
**Contribution:** 2 fair
**Rating:** 5
**Confidence:** 3

**Summary:**

The paper investigates the gradient flow dynamics under the presence of symmetries in the data. Specifically, it compares the dynamics of equivariant MLPs and standard MLPs trained with data augmentation (in the infinite-augmentation limit). It is shown that (1) the sets of stationary points coincide and (2) the dynamics are identical if the initialization is equivariant. However, the stationary points might be unstable for the augmented flow. The theoretical results are supported by controlled numerical experiments.

**Strengths:**

The paper introduces techniques to theoretically analyze different ways of incorporating symmetries in the data. This is a promising research direction, and there appears to be only limited theoretical prior work. The statements seem mathematical sound, and the deduced results are interesting.

**Weaknesses:**

The paper's contribution and practical relevance are constrained by the following limitations:
1. The results only hold for gradient flow.
2. The results on the augmented flow only hold in the "infinite-augmentation limit".
3. It is unclear how the results transfer to other architectures than MLPs. In particular, equivariant models use quite specific architectures in practice.
4. Based on the analysis, a regularization based on $\|A_{\mathcal{E}^\perp}\|$ is proposed to stabilize the augmented flow. However, this practically relevant direction is not further analyzed.
5. Even in the simple (and somewhat artificial) numerical examples, the different behavior for rotations cannot be deduced from the present theory, and only a heuristic argument is presented.

Moreover, it would be good to explain which of the proof techniques are novel and which can already been found in the literature.

Minor comment: For certain citations, it would be good to use `\citep` instead of `\cite` or `\citet`.

**Questions:**

1. "The third assumption guarantees that the loss-landscape is ’unbiased’ towards group transformations, which is
certainly required to train any model respecting the symmetry group". This seems to be a sensible choice but not necessarily required?

2. Remark 2: Why can one move the integral w.r.t. the Haar measure into the loss for an arbitrary loss?

3. "The latter fact is crucial – it will allow us analyse the dynamics of gradient flow." In view of Remark 2, both losses seem to correspond to $R^\mathrm{aug}$, so it is unclear why they have different gradient flow dynamics.

4.  "[...] we use layer normalization Ba et al. (2016) which can be accommodated as an equivariant nonlinearity $\sigma_i$": It would be good to comment on the learnable part of layer normalization.

5. Can some of the paper's conclusions be put into context with the results of empirical papers, e.g., the ones mentioned in the related works section, i.e., Gandikota et al. (2021); Müller et al. (2021); Gerken et al. (2022)?